# Integrating I(I)/I(III) catalysis in reaction cascade design enables the synthesis of *gem*-difluorinated tetralins from cyclobutanols

Joel Häfliger [1], Louise Ruyet [1], Nico Stübke[1], Constantin G. Daniliuc [1] & Ryan Gilmour [1]✉

Partially saturated, fluorine-containing rings are ubiquitous across the drug discovery spectrum. This capitalises upon the biological significance of the native structure and the physicochemical advantages conferred by fluorination. Motivated by the significance of aryl tetralins in bioactive small molecules, a reaction cascade has been validated to generate novel *gem*-difluorinated isosteres from 1,3-diaryl cyclobutanols in a single operation. Under the Brønsted acidity of the catalysis conditions, an acid-catalysed unmasking/fluorination sequence generates a homoallylic fluoride in situ. This species serves as the substrate for an I(I)/I(III) cycle and is processed, via a phenonium ion rearrangement, to an (isolable) 1,3,3-trifluoride. A final C(sp$^3$)-F bond activation event, enabled by HFIP, forges the difluorinated tetralin scaffold. The cascade is highly modular, enabling the intermediates to be intercepted: this provides an expansive platform for the generation of structural diversity.

The development of enabling technologies to generate fluorinated analogues of bioactive leads is a core research endeavor in contemporary catalysis[1–11]. This reflects the clinical importance of fluorination in reconciling physicochemical limitations with promising bioactivity profiles[12,13]. Diversifying the existing drug discovery module portfolio, in a sustainable and atom economic fashion[14,15], has created a fertile ground to advance main group catalysis-based fluorination reactions. In particular, the I(I)/I(III) catalysis manifold[16–19] has proven to be well-suited to this challenge on account of the inexpensive nature of the aryl iodide organocatalyst and the availability of simple organic oxidants and amine•HF reagents[20–23]. More recently, efforts to leverage the intrinsic acidity of the catalysis conditions in multi-step processes have come into focus[24]. Compelling arguments to pursue this research line include (i) circumventing substrate limitations through direct in situ generation, and (ii) the possibility to increase structural complexity in post-catalysis events. Motivated by the prominence of aryl tetralins and fluorinated cycloalkyl motifs in bioactive small molecule discovery (Fig. 1A)[25,26], it was envisaged that this conceptual framework may be advantageous in generating fluorinated analogues. A one-pot

cascade was envisaged in which the direct conversion of 1,3-diarylcyclobutanols to *gem*-difluoro tetralins might be achieved via the merger of Brønsted acid activation and I(I)/I(III) catalysis in a single operation (Fig. 1B). Specifically, it was envisioned that, under the acidic I(I)/I(III) catalysis fluorination conditions with HF, dehydration of the cyclobutanol (**1**) would rupture the ring (**I ↔ II**) and generate the homoallylic fluoride **III** in situ. This would ultimately complement the elegant studies by Lanke and Marek on the generation of *trans*–1,2-disubstituted homoallylic fluorides, via cyclopropylcarbinyl/bicyclobutonium cation formation, from cyclopropyl carbinols[27]. In addition to the well-documented involvement of cyclopropylcarbinyl/bicyclobutonium cations[27–29], direct fluorination of the proposed cyclobutonium species **I** would also account for the generation of homoallylic fluoride **III**. It is pertinent to note that Li and co-workers observed disparate reactivity when exposing aryl-substituted methylene cyclopropanes to Selectfluor® and HF: this triggered a Wagner-Meerwein rearrangement to generate difluorocyclobutanes[30]. In our postulated reaction sequence, the process of in situ substrate formation forges a 1,1-disubstituted alkene: this can then engage in an I(I)/I(III) catalysis cycle[31],

---

[1]Institute for Organic Chemistry, Westfälische Wilhelms-Universität (WWU) Münster, Corrensstraße 36, 48149 Münster, Germany.
✉e-mail: ryan.gilmour@uni-muenster.de

**A. Selected Bioactive Small Molecules**

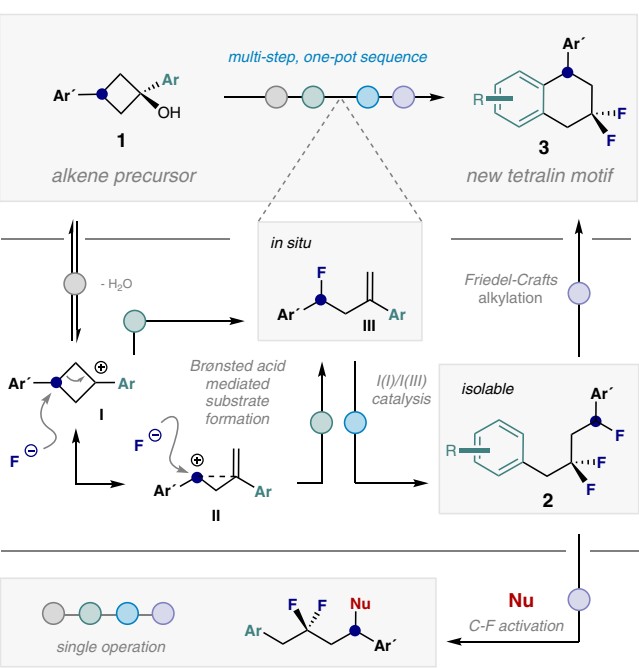

**B. Reaction Design**

**Fig. 1 | Motivation and conceptual framework for the study. A** Selected examples of bioactive small molecules containing the 1-phenyl-1,2,3,4-tetrahydronaphthalene unit. **B** Integrating an I(I)/I(III) catalysis event in a one-pot approach for the synthesis of fluorinated 1-aryl-1,2,3,4-tetrahydronaphthalenes from simple 1,3-diarylcyclobutanols.

enabling a regioselective 1,1-difluorination[32–39] to occur via a precedented phenonium ion rearrangement[40].

Inspired by the seminal work of Paquin and co-workers on HFIP-enabled activation of benzylic C-F bonds[41,42], it was reasoned that a Friedel-Crafts-type cyclisation would furnish the target scaffold and demonstrate the value of integrating I(I)/I(III) catalysis in cascade reaction design. If successful, this would logically lead to an exploration of alternative exogenous nucleophiles, thereby further enhancing the modularity of the paradigm (Fig. 1B, bottom).

## Results and discussion

### Transformation of cyclobutanols (1) to trifluorides (2)

To validate the working hypothesis (Fig. 1B), a process of reaction deconstruction[43] was initiated beginning with the formation of the key trifluoride (*major*–**1c** → **2c**, Table 1). It seemed likely that the Brønsted acidity of the conditions would facilitate dehydration of the 1,3-diarylcyclobutanol *major*–**1c** with concomitant rearrangement, via the transient cation (**I** ↔ **II**), to generate the allylic fluoride motif (**III**). This species would then be intercepted by the I(I)/I(III) catalysis cycle, initiating a phenonium ion rearrangement/fluorination sequence to enable three C(sp³)-F bonds to be forged in a single operation.

To explore the feasibility of this process, substrate **major-1c** (*cis*) was exposed to amine:HF (1:5) (see General Procedure D in the ESI for further information on the preparation of amine:HF mixtures) in CHCl₃ together with *p*-TolI (20 mol%) and Selectfluor® (1.5 eq.) as the catalyst and oxidant, respectively[44]. It is pertinent to mention that crystals of the *trans*-isomer of the starting cyclobutanol (**minor-1c**, CCDC 2239011) could be isolated and subjected to X-ray diffraction analysis (see Table 1 legend). After 18 h at ambient temperature, product **2c** was formed in 74% yield thereby providing confidence in the reaction design (entry 1).

Cognisant of the impact Brønsted acidity has on the regioselectivity of I(I)/I(III)-catalysed fluorination reactions[45], the amine:HF ratio was adjusted stepwise to 1:6.5 (entries 2–4): this allowed the optimal ratio of 1:5.5 to be identified (entry 2, 81% yield by ¹⁹F NMR, 74% isolated yield). Neither changes of solvent (entries 5–8) nor catalyst (entries 9 and 10) led to further enhancements in efficiency. Moreover, a reduction in catalyst loading manifested itself in lower yield (entry 11).

Control reactions in the absence of the HF source (entry 12), *p*-TolI organocatalyst (entry 13), or oxidant (entry 14) were unsuccessful, and support the postulated I(I)/I(III) manifold. To explore the impact of the relative stereochemistry of the substrate on reaction efficiency, a diastereomeric mixture of **1c** enriched with the minor *trans*-isomer (**minor–1c**) was exposed to the standard conditions. As expected, comparable outcomes were observed (entry 2 vs. entry 15).

Having established an optimised catalysis protocol, a series of 1,3,3-trifluorobutanes **2** were prepared from 1,3-diarylcyclobutanols with electronically modulated aryl rings (Fig. 2). During the course of this study, a general trend was noted: for electron rich systems, the amine:HF ratio had to be lowered for optimal efficiency, whereas electron deficient systems required higher amine:HF ratios. For the electron rich phenyl or fluorophenyl substituents, compounds **2a** and **2b** were obtained in 57% and 50%, respectively. Introducing halogens such as chlorine and bromine allowed the formation of the desired trifluorinated products **2c** and **2d** in higher yields (74% and 84% respectively). In the case of the deactivated CF₃ derivative **2e**, extending the reaction time to 42 h was required to generate the product in 67% yield. Next, the effect of varying the R₂ substituent was investigated whilst keeping R₁ = H constant. This enabled the halogenated series **2f**, **2g** and **2h** to be generated as well as the trifluoromethoxy substituted product **2i** (up to 77% yield). Efficient formation of the trifluoromethylated product **2j** and nitrile **2k** could also be realised (67% and 45% respectively) by slightly elevating the amine:HF ratio to 1:6.5. The introduction of a biphenyl substituent (**2l**) and inclusion of *meta*-substituents (**2m**) were also compatible with the protocol. In the case of *ortho*-substituents, extended reaction times were required (e.g. **2n**). In a reversal of circumstances, the impact of modifying R₂ whilst leaving R₁ unchanged was investigated. Cascade processes to furnish the halogenated substrates **2o**, **2p** and **2q** were successful (up to 67%). Moreover, the electron-deficient products **2r** and **2s** could be prepared with comparable efficiency (66% and 67% yield, respectively).

Cognisant of the synthetic utility of aryl bromides for subsequent downstream cross coupling, an additional series with R₁ = Br was explored. Synthetically useful yields were obtained for products **2t** (68%) and **2u** (77%), as well as for the trifluoromethyl derivative **2v** (71%) and triflate **2w** (82%). Further substitution of the cyclobutanol by addition of a methyl group at C3 was tolerated and enabled the trifluoride **2x** to be accessed (57% after 42 h). Finally, the scope of this transformation was found to be compatible with heterocycle-containing substrates as is evident from the 3-phenylpyridine-derivative **2y** (63% yield).

The *bis*-trifluoromethyl derivative **2e** was crystalline and it was possible to unequivocally establish the molecular connectivity created in this cascade by single crystal diffraction (Fig. 3, CCDC 2239010). A slight difference in C-F bond lengths was noted for the aliphatic and

**Table 1 | Optimisation of the transformation of cyclobutanol major−1c (cis) to the ring opened product 2c[a]**

catalyst (20 mol%), Selectfluor® (1.5 eq.)

Amine•HF 1:X (0.5 mL), solvent (0.5 mL), rt, 18 h

*major*-1c (*cis*)
X-ray of *minor*-1c (*trans*) shown below

(±)-2c

| Entry | Solvent | Amine:HF | Catalyst | Yield 2c [%][b] |
|---|---|---|---|---|
| 1 | CHCl₃ | 1:5.0 | pTolI | 74 |
| 2 | CHCl₃ | 1:5.5 | pTolI | 81 (74)[c] |
| 3 | CHCl₃ | 1:6.0 | pTolI | 73 |
| 4 | CHCl₃ | 1:6.5 | pTolI | 62 |
| 5 | DCM | 1:5.5 | pTolI | 69 |
| 6 | DCE | 1:5.5 | pTolI | 72 |
| 7 | MeCN | 1:5.5 | pTolI | <5% |
| 8 | toluene | 1:5.5 | pTolI | 72 |
| 9 | CHCl₃ | 1:5.5 | PhI | 72 |
| 10 | CHCl₃ | 1:5.5 | pMeO-PhI[d] | 39 |
| 11[e] | CHCl₃ | 1:5.5 | pTolI | 69 |
| 12 | CHCl₃ | – | pTolI | <5 |
| 13 | CHCl₃ | 1:5.5 | – | <5 |
| 14[f] | CHCl₃ | 1:5.5 | pTolI | <5 |
| 15[g] | CHCl₃ | 1:5.5 | pTolI | 76 |

[a]Standard reaction conditions: *major*-1c (0.2 mmol), Selectfluor® (1.5 equiv.), amine•HF mixture (0.5 mL), solvent (0.5 mL), catalyst (20 mol%), 18 h, rt. The relative configuration of the starting material was determined by comparison with the X-ray structure of *minor*-1c (CCDC 2239011, see ESI for further information). Thermal ellipsoids are shown at 50% probability. Only one molecule of two found in the asymmetric unit is shown.
[b]Determined by ¹⁹F NMR using ethyl 2-fluoroacetate as internal standard. Isolated yield in parentheses.
[c]NMR- and isolated yields are reported as the average of two independent experiments.
[d]4-Iodoanisole.
[e]10 mol% of catalyst was used.
[f]No Selectfluor® was used.
[g]An enriched mixture of starting material favouring the *minor*-1c diastereoisomer (d.r. 89:11) was used.

benzylic environments (1.377 and 1.376 Å versus 1.396 Å, respectively)[46].

## Transformation of cyclobutanols (1) to *gem*-difluorinated tetralins (3)

In the course of the optimisation of the 1,3,3-trifluorination reaction of *major*−1c, traces of the cyclised Friedel-Crafts product 3c were detected when the reactions were conducted at higher amine:HF ratios. This preliminary validation of the one-pot, multi-step conversion of *major*-1c → 3c provided an excellent foundation for reaction development (Table 2, entry 1). To identify an appropriate amine:HF mixture for benzylic fluoride activation, the impact of systematically adding Olah´s reagent and CHCl₃ was investigated.

Gratifyingly, the addition of solvent and Olah´s reagent (0.75 and 1 mL each) yielded notable quantities of the desired 3,3-difluorotetrahydronaphthalene 3c (entries 2 and 3). Reducing the volume of CHCl₃ had a beneficial impact on reaction efficiency, and its exclusion enabled the product to be generated in 72% from starting cyclobutanol *major*-1c (entry 5). Inspired by the seminal work of Paquin and co-workers on the activation of benzylic C-F bonds[41,42], HFIP was investigated as a substitute for additional Olah´s reagent[47,48]. In this case, the

addition of a mixture of HFIP and CHCl₃ (2 mL, 1:1) led to the formation of desired product after 24 h (entry 6). Once again, eliminating the CHCl₃ had a beneficial impact on the yield of 3c (55%, entry 7). Moreover, increasing the amount of HFIP to 2 mL led to higher yields (entry 8, 70% isolated yield): this is comparable to the yields attained using additional Olah´s reagent (entry 5, 72%). The advantage of this direct, one-pot protocol was immediately apparent following a comparison with the stepwise synthesis: this led to product 3c being generated in 70% and 48%, respectively (entry 9, see Stepwise Synthesis of 3c from *major*-1c in the ESI for further details).

To determine the scope and limitations of this fluorinative cascade to generate the target tetralins, a set of 1,3-diarylcyclobutanols were exposed to the standard conditions (Fig. 4). The reaction proved to be compatible with phenyl- and fluorophenyl substituents (3a and 3b), and halogenated substrates were particularly well-suited, enabling diversely halogenated scaffolds 3c and 3d to be generated (up to 76% yield). This latter observation is in line with the observations described in Fig. 2. Subsequently, R₁ was varied whilst R₂ remained constant (R₂ = H). This one pot protocol enabled halogenated derivatives 3f, 3g and 3h to be forged, as well as the trifluoromethoxy species 3i. The inclusion of electron-rich biphenyl substituents (3l), as well as *meta*- and *ortho*-

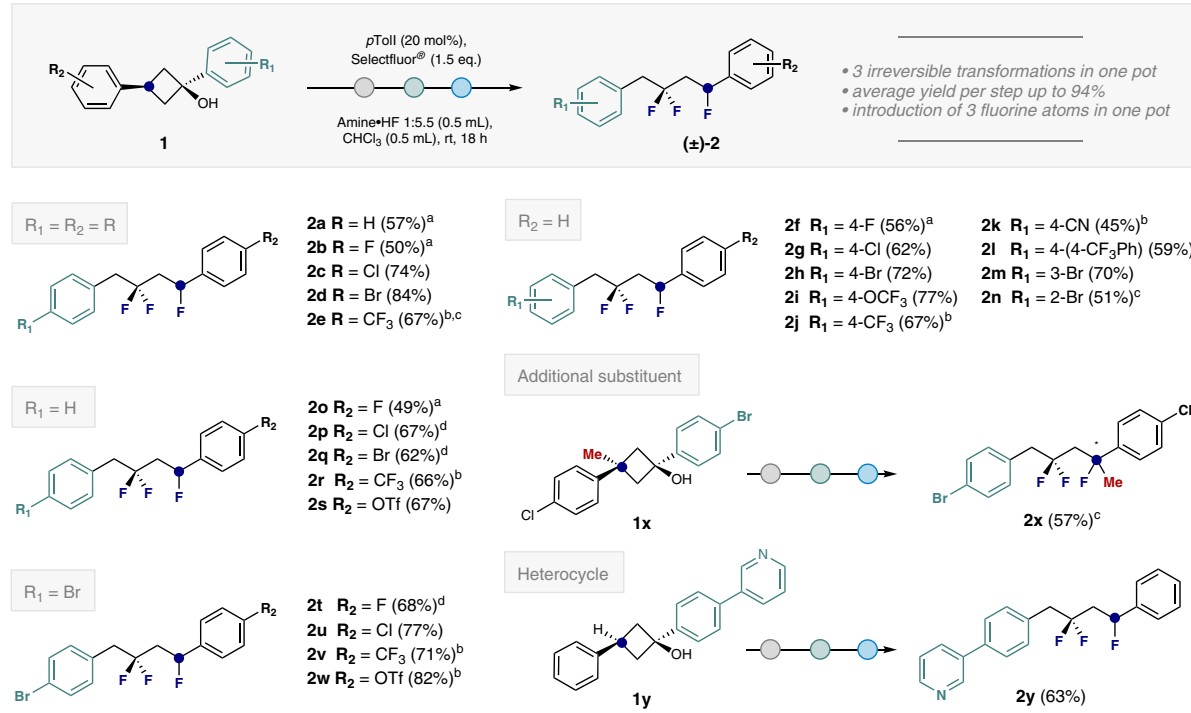

**Fig. 2 | Scope for the trifluorination of 1,3-diarylcyclobutanol derivatives 1.** Isolated yields are given in parentheses. Where possible, substrate **1** was used as a single diastereoisomer. See ESI for full details. [a] An amine:HF ratio of 1:4.5 was used. [b] An amine:HF mixture with a ratio of 1:6.5 was used. [c] Reaction stirred for 42 h. [d] An amine:HF ratio of 1:5.0 was used.

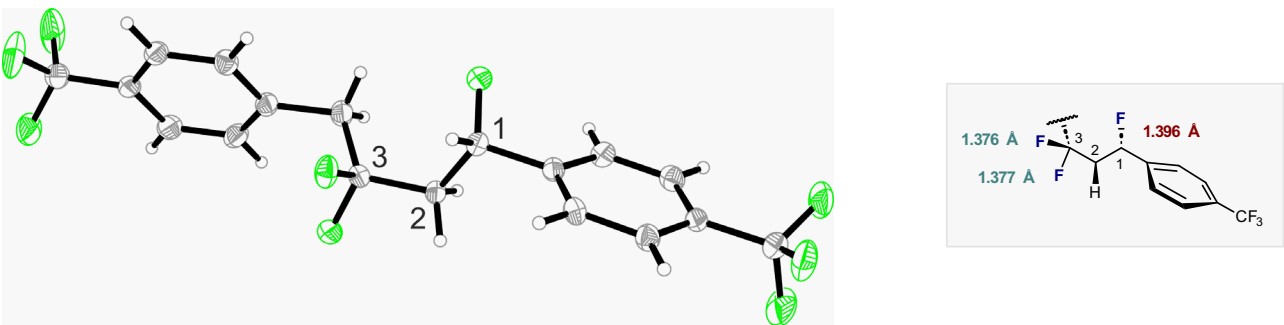

**Fig. 3 | Crystal structure analysis of nonafluoride 2e (CCDC 2239010).** The main conformation found in the asymmetric unit (80%) is represented. Thermal ellipsoids are shown at 50% probability.

substitution patterns (**3m** and **3n**), proved to be compatible with the reaction conditions. It is interesting to note that in the case of **3m**, only the 7-Br regioisomer was isolated, presumably to mitigate destabilising non-bonding interactions. Next, the impact of varying $R_2$ on reaction efficiency was explored. Gratifyingly, the halogenated tetralin series **3o**, **3p** and **3q** were obtained efficiently (up to 62% yield). It was possible to generate the triflate **3s** (56% yield) but required an extension of the reaction time to 72 h. Switching $R_1$ to the valuable bromide handle enabled products **3t** and **3u** to be prepared in up to 71% yield.

Pushing the limits of the process revealed electron-deficient substrates to be challenging. However, the desired tetralins could be generated by a two-step compromise and the use of Olah´s reagent for the final activation/Friedel-Crafts cyclisation (Fig. 4, bottom). To that end, the isolated intermediates (**2**) were treated with a mixture of Olah´s reagent and CHCl₃ (1:1) and stirred for 24 h at the specified temperature. Leveraging this platform, it was possible to access the bis- and mono-CF₃ species **3e**, **3j**, **3r** and **3v** (up to 98% yield). To enable further functionalisation, tetralin **3w** bearing orthogonal C(sp²)-Br and

C(sp²)-OTf motifs was synthesised in 75%. Finally, the pyridine-containing tetralin **3y** was accessed by this protocol in 68%.

## Synthetic applications

The efficiency of the C-F bond activation, coupled with the modest nucleophilicity of the aryl rings in this study, provided an opportunity to expand the scope of the process by introducing superior, external nucleophiles. To provide preliminary validation of this notion, AcOH, MeOH and *p*-xylene were introduced to the one-pot cascade reaction with ***major*-1c** (*cis*) (Fig. 5). All three transformations successfully generated the desired products in up to 72% yield. The oxidative lability of many nucleophiles called for the development of a complementary stepwise process to encompass *S*- and *N*-based nucleophiles (**7**, **9** and **10**). By treating **2c** with 2-mercaptobenzothiazole in a mixture of HFIP:CHCl₃ (3:1) at 40 °C for 66 h, thioether **7** was obtained in 86% yield. Facile oxidation of **7** with *m*-CPBA furnished the olefination precursor, sulfone **8**. A similar strategy enabled the fluorinated thioether **9** to be forged in 93% yield. Amination was achieved using

**Table 2 | Optimisation of the transformation of cyclobutanol *major*-1c to the cyclised product 3c[a]**

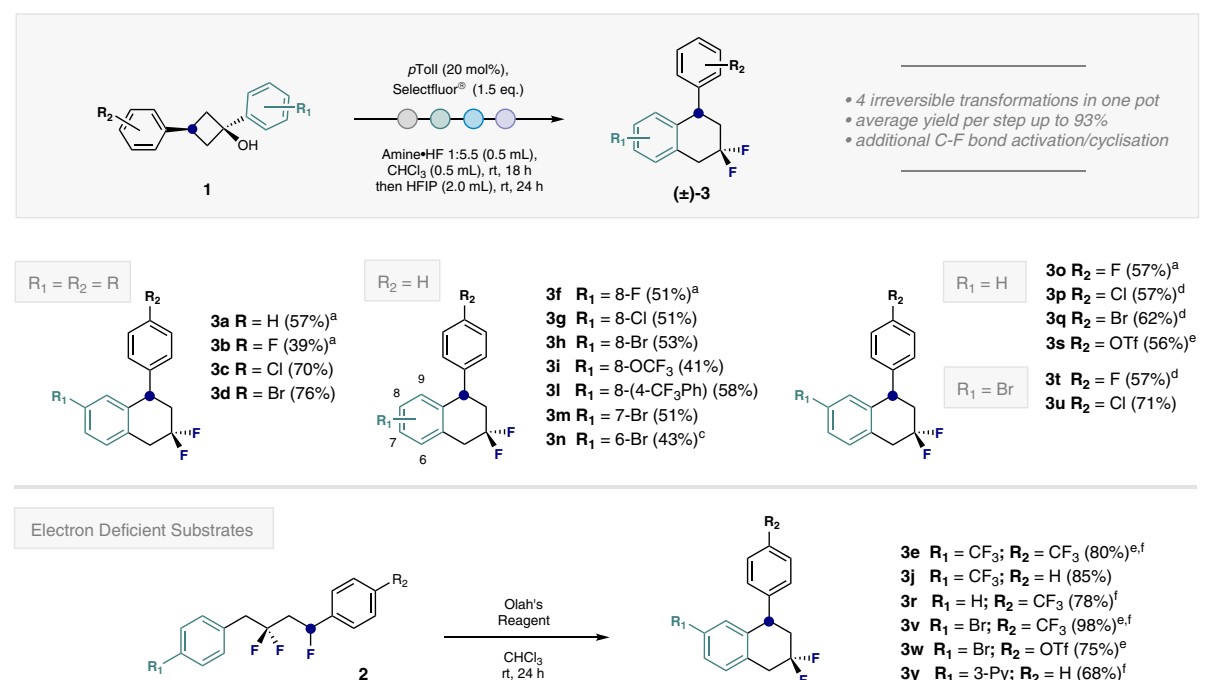

| Entry | Solvent (mL) | Brønsted Acid (mL) | Yield 3c [%][b] |
|---|---|---|---|
| 1[c] | – | – | traces |
| 2 | CHCl₃ (0.75) | Py•HF (0.75) | 57 |
| 3 | CHCl₃ (1.0) | Py•HF (1.0) | 68 |
| 4 | CHCl₃ (0.5) | Py•HF (1.0) | 70 |
| 5 | – | Py•HF (1.0) | 72 |
| 6 | CHCl₃ (1.0) | HFIP (1.0) | 13 |
| 7 | – | HFIP (1.0) | 55 |
| 8 | – | HFIP (2.0) | 71 (70) |
| 9[d] | – | HFIP (2.0) | (48) |

[a]Standard reaction conditions: Intermediate **2c** was prepared according to the procedure described in Table 1 on a 0.2 mmol scale. Additional solvent and/or Brønsted acid was added after 18 h. The reaction was stirred for an additional 24 h at rt.

[b]Determined by ¹⁹F NMR using ethyl 2-fluoroacetate as internal standard. Isolated yield in parentheses.

[c]**2c** was prepared with an amine:HF ratio of 1:6.5. No solvent nor acid activator was added after 18 h.

[d]**3c** was prepared stepwise, isolating **IIIc** and **2c**. The stepwise synthesis was performed on a 0.5 mmol scale. Full details provided in the ESI.

**Fig. 4 | Scope for the trifluorination/cyclisation sequence.** Isolated yields are given in parentheses. Where possible, substrate **1** was used as a single diastereoisomer. See ESI for full details. [a]An amine:HF mixture with a ratio of 1:4.5 was used.

[b]An amine:HF ratio of 1:6.5 was used. [c]For the formation of intermediate **2n**, the reaction time was extended to 42 h. [d]An amine:HF ratio of 1:5.0 was used. [e]The reaction was stirred for 72 h. [f]The reaction was conducted at 40 °C.

acetonitrile as the exogenous nucleophile[49], where a subsequent Ritter-type reaction liberated the γ,γ-difluoroamine **10** in 78% yield.

To demonstrate the synthetic utility of the fluorinated tetralins in the arenas of contemporary medicinal chemistry and organic materials design, selected synthetic modifications were conducted (Fig. 6). Given the importance of tetralins in drug discovery, a short synthesis of the difluorinated analogue of Nafenopin was executed from triflate

**3s** (Fig. 6A). Saponification using NEt₄OH enabled phenol **11** to be prepared and processed to the target **12** via an alkylation / deprotection sequence.

Product **12** was crystalline and it was possible to determine the structure via X-ray analysis (CCDC 2239012). The near perfect half-chair of the central ring system further underscores the effectiveness of the *gem*-difluoro motif as an isosteric replacement for methylene

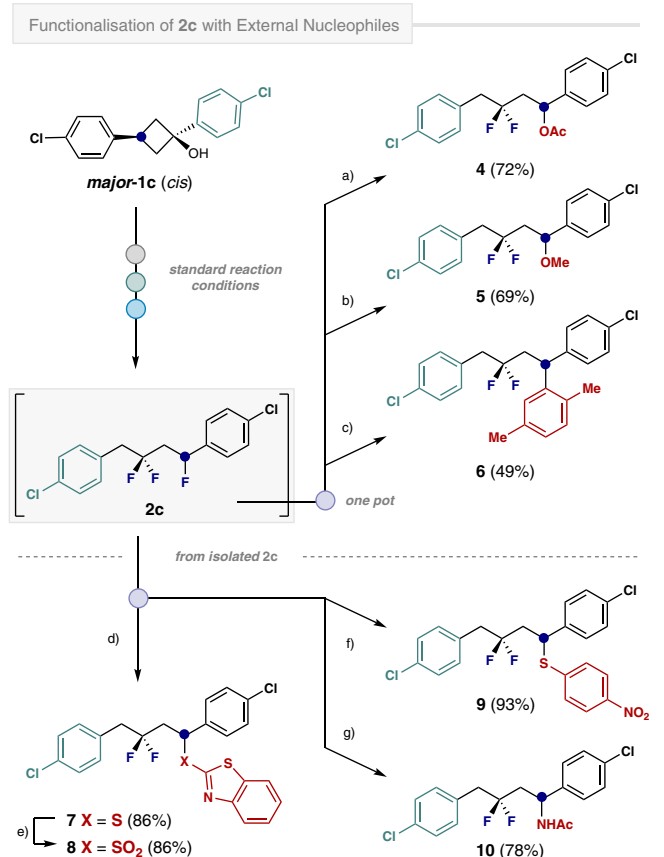

Functionalisation of **2c** with External Nucleophiles

**4** (72%)

**5** (69%)

**6** (49%)

*one pot*

**2c**

*from isolated 2c*

**7 X = S** (86%)

**8 X = SO₂** (86%)

**9** (93%)

**10** (78%)

**Fig. 5 | Synthetic modifications of trifluoride 2c.** Isolated yields in parentheses. One-pot reactions: **2c** was prepared under standard reaction conditions: a) AcOH (1.0 mL), Olah's reagent (1.0 mL), 24 h, rt. b) MeOH (1.0 mL), Olah's reagent (2.0 mL), 48 h, rt. c) *p*-xylene (1.0 mL), Olah's reagent (1.0 mL), 48 h, rt. Reaction conditions using isolated **2c**: d) **2c** (0.2 mmol), 2-mercaptobenzothiazole (0.4 mmol), HFIP (1.2 mL), CHCl₃ (0.4 mL), 66 h, 40 ˚C. e) **7** (0.2 mmol), *m*-CPBA (0.44 mmol), CHCl₃ (3.0 mL), 15 h, −10 ˚C to rt. f) **2c** (0.2 mmol), 4-nitrobenzenethiol (1.0 mmol), HFIP (1.2 mL), CHCl₃ (0.4 mL), 24 h, 40 ˚C. g) **2c** (0.2 mmol), Olah's reagent (1.0 mL), MeCN (1.0 mL), 20 h, 40 ˚C.

A. Synthesis and Crystal Structure of Nafenopin-Derivative **12**

**11** (69%)

**12** (58% over 2 steps)

**3s**

B. Synthetic Modifications of **3q**

*Sonogashira*

**13** (93%)

*Suzuki-Miyaura*

**14** (83%)

*Ullmann*-type C-N-coupling

**15** (97%)

**3q**

**Fig. 6 | Synthetic modifications of 3s and 3q. A** Synthesis and crystal structure of nafenopin derivative **12**. Reaction conditions: a) **3s** (0.58 mmol), NEt₄OH 10 wt% in H₂O (1.16 mmol), 1,4-dioxane (1.75 mL), 90 min, rt. b) **11** (0.30 mmol), *tert*-butyl 2-bromo-2-methylpropanoate (1.50 mmol), MgSO₄ (0.30 mmol), K₂CO₃ (1.20 mmol), DMF (2.0 mL), 24 h, 100 ˚C. c) TFA (6.0 mmol), DCM (3 mL), 1 h, 0 ˚C to rt. Crystal structure of compound **12** (CCDC 2239012) showing the main conformation found in the asymmetric unit (84%). Thermal ellipsoids are shown at 50% probability. **B** Synthetic modifications of tetralin **3q**. d) **3q** (0.20 mmol), ethynyl-triisopropylsilane (0.30 mmol), Pd(PPh₃)₂Cl₂ (0.02 mmol), CuI (0.04 mmol), NH(*i*-Pr)₂ (0.5 mL), 14 h, 70 ˚C. e) **3q** (0.20 mmol), benzo[d][1,3]dioxol-5-ylboronic acid (0.22 mmol), Pd(PPh₃)₄ (0.005 mmol), PPh₃ (0.02 mmol), Na₂CO₃ (0.24 mmol), EtOH/H₂O (5:1, 0.72 mL), 14 h, 80 ˚C. f) **3q** (0.20 mmol), indole (0.20 mmol), CuI (0.02 mmol), *N,N'*-dimethyldiaminoethane (0.08 mmol), K₃PO₄ (0.42 mmol), toluene (0.5 mL), 20 h, 110 ˚C. Isolated yields in parentheses.

groups. Finally, to demonstrate the suitability of a representative tetralin in the generation of carbon rich scaffolds, Sonogashira (**13**, 93%), Suzuki (benzodioxol **14**, 83%) and Ullmann-type coupling (**15**, 97%) were successfully conducted (Fig. 6B).

Hypervalent iodine catalysis has significantly augmented the fluorination portfolio, enabling the direct installation of C(sp³)-F bonds in alkene substrates without the need for substrate pre-functionalisation. The fusion of this platform with simple HF sources confers an array of advantages for reaction design, not least the ability to leverage the intrinsic acidity of the reaction medium to unmask substrates in situ. In this study, the dehydration of easily accessible cyclobutanols has been validated as a platform to trigger a fluorinative skeletal rearrangement cascade to access biologically relevant aryl tetralins in a highly regioselective fashion. Subsequent benzylic fluorination forges a (1,1-disubstituted) styrenyl substrate that can be further intercepted by an I(I)/I(III) catalysis manifold. Activation of the alkene by the ephemeral hypervalent iodine centre triggers a phenonium ion rearrangement, ultimately generating the *gem*-difluoromethyl unit, which forms part of a trifluoro motif. This motif can be isolated or processed, via C(sp³)-F bond activation, to the desired tetralin. Moreover, by introducing a competitive exogenous (*C*- and *O*-based) nucleophile, it is possible to override the intramolecular process to further broaden the modularity of the process.

## Methods

### General procedure for the synthesis of (±)−2

Cyclobutanol derivative **1** (0.20 mmol, 1.0 eq.) and *p*-TolI (8.7 mg, 0.04 mmol, 20 mol%) were dissolved in CHCl₃ (0.5 mL) in a Teflon®-vial (5 mL total volume). Subsequently, NEt₃·3HF and Olah's reagent were added with the appropriate ratio (0.5 mL total volume, for more information, see previous publication of this group[36]). Finally, Selectfluor® (106 mg, 0.3 mmol, 1.5 eq.) was added to the reaction mixture in one portion. The reaction mixture was stirred for 18 h at room temperature. The reaction mixture was diluted with DCM (2 mL) and poured in saturated aqueous NaHCO₃ (100 mL). The aqueous layer was extracted with DCM (3 × 30 mL). The combined organic layers were dried over Na₂SO₄ and the solvent was removed under reduced pressure. The crude product was purified by flash column chromatography.

## Caution

Olah's reagent is highly toxic and corrosive. Direct exposure should be avoided. In the case of skin exposure, immediate treatment of the affected skin area with calcium gluconate gel is necessary to prevent serious chemical burns.

## General procedure for the synthesis of (±)−3 (one pot procedure)

The 1,1,3-trifluorobutane derivative (±)−2 was prepared according to the general procedure on a 0.200 mmol·scale with the indicated amine•HF mixture. Instead of quenching the reaction after 18 h, HFIP (2.0 mL) was added and the reaction was stirred for another 24 h at room temperature. The reaction mixture was diluted with DCM (2 mL) and poured in saturated aqueous NaHCO$_3$ (100 mL). The aqueous layer was extracted with DCM (3 × 30 mL). The combined organic layers were dried over Na$_2$SO$_4$ and the solvent was removed under reduced pressure. The crude product was purified by flash column chromatography.

## Caution

Olah's reagent is highly toxic and corrosive. Direct exposure should be avoided. In the case of skin exposure, immediate treatment of the affected skin area with calcium gluconate gel is necessary to prevent serious chemical burns.

## General procedure for the synthesis of (±)−3 (stepwise procedure)

Isolated 1,1,3-trifluorobutane derivative (±)−2 (0.200 mmol, 1.0 eq.) was dissolved in CHCl$_3$ (1.0 mL). Olah´s reagent (1.0 mL) was added and the reaction was stirred at the indicated temperature for the indicated time. The reaction mixture was diluted with DCM (2 mL) and poured in saturated aqueous NaHCO$_3$ (200 mL). The aqueous layer was extracted with DCM (3×30 mL). The combined organic layers were dried over Na$_2$SO$_4$ and the solvent was removed under reduced pressure. The crude product was purified by flash column chromatography.

## Caution

Olah's reagent is highly toxic and corrosive. Direct exposure should be avoided. In the case of skin exposure, immediate treatment of the affected skin area with calcium gluconate gel is necessary to prevent serious chemical burns.

## Data availability

CCDC 2239011 contains the supplementary crystallographic data for compound *minor*-1c (*trans*). CCDC 2239010 contains the supplementary crystallographic data for compound 2e. CCDC 2239012 contains the supplementary crystallographic data for compound 12. The data can be obtained free of charge from The Cambridge Crystallographic Data Centre [http://www.ccdc.cam.ac.uk/data_request/cif]. Supplementary Information is available for this paper. All data are available in the main text or the supplementary materials. Correspondence and requests for materials should be addressed to Prof. Ryan Gilmour (ryan.gilmour@uni-muenster.de).

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

## Acknowledgements

We gratefully acknowledge the support provided by the technical departments of the Institute for Organic Chemistry at the WWU Münster. We acknowledge financial support from the WWU Münster (R.G.), the Deutsche Forschungsgemeinschaft (SFB 858, R.G.) and the European Research Council (ERC Consolidator Grant RECON 818949, R.G.).

## Author contributions

Initial project idea: J.H., R.G., Conceptualization: J.H., R.G., Methodol-ogy: J.H., R.G., Investigation: J.H., L.R., N.S., C.D., Funding acquisition: R.G., Project administration: R.G., Supervision: J.H., L.R., R.G., Writing: J.H., L.R., R.G.

## Funding

## Competing interests

The authors declare no competing interests.
