## [Peer Review File · Nature Communications]

REVIEWER COMMENTS

Reviewer #1 (Remarks to the Author):

The authors report the synthesis and characterization of gem-Difluorinated Tetralins from cyclobutanols using I(I)/I(III) mediated catalyst.

The partially fluorinated scaffolds indeed offer scope for the future development and applications of these molecules in the drug industry, keeping in mind that molecules containing organic fluorine are of significance. But what is missing in this paper is whether these molecules have been screened for improved bioefficacy or not. Is the role of organic fluorine established in comparison to the non-fluorinated analogues of these molecules.

Even if the authors have not done these measurements or studies in the current work, I encourage them to undertake such studies to unequivocally establish their claim that these indeed are ubiquitous in the drug discovery spectrum.

A careful look at the crystal structures and the checkCIFs reveal that the data/parameter ratio was poor. The accepted ratio is 8-10. Is there any specific reason for this. Is this related to the morphology of the crystals.

Also there exists disorder in the crystal structure. It will be of relevance if the authors mention the ratio of the two independent conformations in the ESI wherein details on data collection are mentioned.

Also in Fig 2 and remaining places, the ORTEP view depicting the thermal ellipsoids is required as that represents a more realistic view of the atoms in molecules in crystals. Hence the ellipsoidal view from Mercury/any related software must be made and presented.

Overall, I congratulate the authors on this excellent piece of work in synthesis and characterization of organo-fluorine compounds. Also, the authors can look at the polymorphic events associated with these compounds and explore the role of weak C-H...F interactions in the crystal packing of these molecules.

Keeping in mind the fact that the above-mentioned corrections will be incorporated, I support publication of this manuscript in Nature Communications.

Reviewer #2 (Remarks to the Author):

This really is a striking paper which explores methodology exposing really quite exotic structural space. It starts from a slightly unusual but synthetically accessible class of cyclobutanol rings and in a single protocol generates the trifluorinated products 2, and generally in very good yield. This transformation is widely exemplified.

The reaction utilises the in-situ generation of fluoroiodinanes, as electrophiles. This is not an intuitive transformation, and that is what makes the contribution most impressive, but it evolves from a body of work in the Gilmour lab using these reagents, and understanding their mechanism. The process involves carbocation rearrangement, nucleophilic fluorinations with concomitant aryl migration. The versatility is fully explored here and to exciting effect. The reagents are cheap and available and the chemistry appears to be robust. Compounds 2 are then further processed to tetralins containing a geminal difluoromethylene group. This is an attractive ring system in medchem, and the approach introduces fluorine which is well known to have subtle effect on pharmacokinetics. The transformations to tetralones is also widely exemplified, and in one case an analogue of nafenopin is prepared with apparent ease.

The benzylic fluorine in compounds 2c are readily activated for substitution with O, S, N and C nucleophiles, again offering access to a very wide range of products, and these products are all compatible too with a range of Pd cross-coupling reactions, which offers further access to considerable diversification.

There is clear elegance in this work, and it will be attractive to those involved in bioactives discovery.

I only have minor comments. The isomer trans 1c is mentioned in the text, and the minor isomer of 1c is mentioned in Table 1 legend, but it is not clear what is the structure of cis and what is the structure of trans. Is cis 1c the structure shown in the Scheme? That could be clarified.

The word 'Gratifyingly' is used four times in the text, this frequency could be significantly reduced.

The paper is concise and impressive in its ingenuity. I am very happy to recommend publication.

Reviewer #3 (Remarks to the Author):

This manuscript by Gilmour and coworkers describes a multi-step sequence starting from diarylcyclobutanols and furnishing fluorinated aryl-tetralins. This one-pot reaction involves acid-catalyzed SN1 of the alcohol and trapping of the carbocation intermediate with fluoride,

difluorination of the alkene with phenonium rearrangement, and C-F activation leading to Friedel-Crafts cyclization. Individually, each step is known, although their combination together is innovative and non-trivial.

- Homoallylic fluorination of cyclopropylcarbanyl and/or bicyclobutonium and/or cyclobutyl cations (Marek et al, J. Am. Chem. Soc. 2020, 142, 5543-5548, which is not cited here.)

- Hypervalent iodine-catalyzed fluorination of alkenes, from this paper's and other authors (references 28-36)

- C-F activation of benzylic fluorides with HFIP for Friedel-Crafts (Paquin, reference 38)

The novel sequence reported here does allow the synthesis of complex difluorinated tetralins. Some discussion about whether this methodology is preferable to deoxofluorination of the corresponding tetralone would be warranted in order to assess how likely it is that the synthetic approach described here is better to access such compounds. The functional group tolerance of the current method is moderate (mostly halogens) but yields are good considering the one-pot sequence of three chemical transformations. A mention of what functional groups are not tolerated in this reaction would be helpful to assess its limitations. Nonetheless, the authors demonstrate its use for the synthesis of a derivative of nafenopin, which showcases the potential use of this method. Of note, the Supporting Information file is nicely organized and presented. The amount and quality of the NMR characterization data is impressive, and the authors are commended for their work.

This paper should be publishable in Nature Communications after minor revisions. Of particular importance, a proposed mechanism with detailed intermediate structures (including rearrangement steps) is needed, either as the reaction design scheme or as a final scheme. This reviewer is particularly skeptical about the proposed nature of the cyclobutyl carbocation intermediate and why it would lead to a homoallylic fluoride product. Creary (J. Org. Chem. 2020, 85, 7086-7096.) has shown that when stabilized cyclobutyl electrophiles are ionized, cyclobutyl and bicyclobutonium cations are formed but eventually provide cyclobutane/cyclobutene products, not homoallylic products. On the other hand, Champagne (10.26434/chemrxiv-2022-h9bq3-v2) has shown that in the Marek system cyclopropylcarbanyl cations are formed and can lead to homoallylic products directly or through homoallylic cations. The latter were always higher in energy, even when the homoallylic position is tertiary and benzylic. As such, this reviewer would hypothesize that in the current system, ionization of cyclobutanols 1 eventually leads to a cyclopropylcarbanyl cation (either directly or through a cyclobutyl/bicyclobutonium intermediate), which would explain the formation of the homoallylic fluoride III without requiring the homoallylic cation II. Some discussion about these possibilities is lacking here.

Minor comments:

- Compounds are all racemic, could it be indicated as so in the schemes?

- That Olah's reagent is $\text{Py}\cdot\text{HF}$ is not indicated anywhere in the manuscript. Also, how the amine $\cdot\text{HF}$ mixtures are made using $\text{Et}_3\text{N}\cdot 3\text{HF}$ and $\text{Py}\cdot\text{HF}$ should be discussed somewhere in the text.

- The impact of solvent quantities should probably be discussed as a function of solute concentration rather than solvent volume (0.5 – 2mL), to make it more general.

REVIEWER COMMENTS

- **Reviewer #1 (Remarks to the Author):**

The authors report the synthesis and characterization of gem-Difluorinated Tetralins from cyclobutanols using I(I)/I(III) mediated catalyst. The partially fluorinated scaffolds indeed offer scope for the future development and applications of these molecules in the drug industry, keeping in mind that molecules containing organic fluorine are of significance. But what is missing in this paper is whether these molecules have been screened for improved bioefficacy or not. Is the role of organic fluorine established in comparison to the non-fluorinated analogues of these molecules. Even if the authors have not done these measurements or studies in the current work, I encourage them to undertake such studies to unequivocally establish their claim that these indeed are ubiquitous in the drug discovery spectrum.

Author Response: We thank the referee for his/her/their very generous assessment of the work and for supporting publication in *Nature Communications*. The referee has raised a very important point regarding the product motifs in drug discovery campaigns. Although we feel that a full biological study in beyond the scope of this methodology investigation, we can disclose that we are working with a lead discovery group to incorporate selected motifs into their module library. Regrettably, we are prohibited from disclosing these findings for proprietary reasons at this time. In Scheme 4 we demonstrated that the compound **3s** could be processed to a fluorinated analogues of nafenopoin to demonstrate that the fluorinated tetralins reported are compatible with the types of processes typically leveraged in medicinal chemistry.

A careful look at the crystal structures and the checkCIFs reveal that the data/parameter ratio was poor. The accepted ratio is 8-10. Is there any specific reason for this. Is this related to the morphology of the crystals.

Author Response: This is an excellent point. We agree with the referee that the data/parameter ratio should be, in general, higher than 8-10 for a quality structure determination. Unfortunately, for compound **2e** and **12** this ratio is slightly below the value of 8 (7.05 for compound **2e** and 7.65 for compound **12**). The observed ratio can be rationalised by various factors which include crystal size (very small, plate-like crystals with small size dimensions - 0.040 mm for **2e**; 0.074/0.096 mm for **12**), light atom structures (C, H, O, F) with poor diffraction scattering at large theta angles and large disordered parts of these molecules (70% for compound **2e** and 68% for compound **12**). We have modified the ESI to include a larger discussion of the X-ray crystal structures (*vide infra*).

Also there exists disorder in the crystal structure. It will be of relevance if the authors mention the ratio of the two independent conformations in the ESI wherein details on data collection are mentioned.

Author Response: We thank the referee for this helpful suggestion. The corresponding ratio of the two-independent conformations for the both disordered structures (compounds **2e** and **12**) have been added to the supporting information and additional information has been added to the manuscript figure legends. For compound **2e** this ratio was found to be 80:20 and for compound **12** 84:16, respectively. The XP pictures of the two conformers for compound **2e** and **12** have also been added to the SI.

Also in Fig 2 and remaining places, the ORTEP view depicting the thermal ellipsoids is required as that represents a more realistic view of the atoms in molecules in crystals. Hence the ellipsoidal view from Mercury/any related software must be made and presented.

Author Response: For all three compounds, the corresponding ORTEP views depicting the thermal ellipsoids with a probability of 50% have been added to the manuscript and ESI. We thank the referee for this helpful suggestion.

Overall, I congratulate the authors on this excellent piece of work in synthesis and characterization of organo-fluorine compounds. Also, the authors can look at the polymorphic events associated with these compounds and explore the role of weak C-H...F interactions in the crystal packing of these molecules. Keeping in mind the fact that the above-mentioned corrections will be incorporated, I support publication of this manuscript in *Nature Communications*.

Author Response: The referee is quite right that an array of interactions can be observed in the packing diagrams. We have added depictions of the three compounds to the supporting information which show weak C-H...F, F...F,

C-H \cdots pi or O-H \cdots O interactions. Again, we wish to thank the referee for the detailed and insightful report. Aside from the ongoing biological work, which we feel is beyond the scope of this methodology study and will be disclosed in due course, we hope that the comments and suggestions from referee 1 have been fully addressed.

- **Reviewer #2 (Remarks to the Author):**

This really is a striking paper which explores methodology exposing really quite exotic structural space. It starts from a slightly unusual but synthetically accessible class of cyclobutanol rings and in a single protocol generates the trifluorinated products 2, and generally in very good yield. This transformation is widely exemplified. The reaction utilises the in-situ generation of fluoriodinanes, as electrophiles. This is not an intuitive transformation, and that is what makes the contribution most impressive, but it evolves from a body of work in the Gilmour lab using these reagents, and understanding their mechanism. The process involves carbocation rearrangement, nucleophilic fluorinations with concomitant aryl migration. The versatility is fully explored here and to exciting effect. The reagents are cheap and available and the chemistry appears to be robust. Compounds 2 are then further processed to tetralins containing a geminal difluoromethylene group. This is an attractive ring system in medchem, and the approach introduces fluorine which is well known to have subtle effect on pharmacokinetics. The transformations to tetralones is also widely exemplified, and in one case an analogue of nafenopin is prepared with apparent ease.

The benzylic fluorine in compounds 2c are readily activated for substitution with O, S, N and C nucleophiles, again offering access to a very wide range of products, and these products are all compatible too with a range of Pd cross-coupling reactions, which offers further access to considerable diversification.

There is clear elegance in this work, and it will be attractive to those involved in bioactives discovery.

I only have minor comments. The isomer trans 1c is mentioned in the text, and the minor isomer of 1c is mentioned in Table 1 legend, but it is not clear what is the structure of cis and what is the structure of trans. Is cis 1c the structure shown in the Scheme? That could be clarified. The word 'Gratifyingly' is used four times in the text, this frequency could be significantly reduced. The paper is concise and impressive in its ingenuity. I am very happy to recommend publication.

Author Response: We are most grateful to the referee for this very supportive summary of the work and for recommending publication in *Nature Communications*. Two corrections were requested and both have been incorporated into the revised version of the manuscript. Specifically, the starting material isomers have been designated as "**major-1c**" (*cis*) and "**minor-1c**" (*trans*). We have also measured the X-ray structure of the minor (*trans*) isomer and added it to the Table 1 to justify the stereochemical descriptors and to also to support the mechanistic discussion (please see referee report #3). The CCDC number has also been included in the text (CCDC 2239011 **minor-1c**).

Finally, the excessive use of the word "Gratifyingly" has been addressed and the term now only appears once in the manuscript. Once again, we thank the expert referee for the positive and supportive comments and helpful suggestions.

- **Reviewer #3 (Remarks to the Author):**

This manuscript by Gilmour and coworkers describes a multi-step sequence starting from diarylcyclobutanols and furnishing fluorinated aryl-tetralins. This one-pot reaction involves acid-catalyzed SN1 of the alcohol and trapping of the carbocation intermediate with fluoride, difluorination of the alkene with phenonium rearrangement, and C-F activation leading to Friedel-Crafts cyclization. Individually, each step is known, although their combination together is innovative and non-trivial.

- Homoallylic fluorination of cyclopropylcarbanyl and/or bicyclobutonium and/or cyclobutyl cations (Marek et al, J. Am. Chem. Soc. 2020, 142, 5543-5548, which is not cited here.)

- Hypervalent iodine-catalyzed fluorination of alkenes, from this paper's and other authors (references 28-36)

- C-F activation of benzylic fluorides with HFIP for Friedel-Crafts (Paquin, reference 38)

Author Response: We thank the referee for the very supportive and insightful comments regarding the study, and the excellent suggestions on how to strengthen it. The referee is absolutely correct that the highly relevant study from Marek and co-workers was not cited in the initial submission. This was an unintentional oversight for which we apologise unreservedly. The introduction has been modified and now contains a sentence to draw attention to the work:

"This would ultimately complement the elegant studies by Lanke and Marek on the generation of trans-1,2-disubstituted homoallylic fluorides, via cyclopropinyl carbocations, from cyclopropyl carbinols.[27]"

The novel sequence reported here does allow the synthesis of complex difluorinated tetralins. Some discussion about whether this methodology is preferable to deoxyfluorination of the corresponding tetralone would be warranted in order to assess how likely it is that the synthetic approach described here is better to access such compounds.

Author Response: Again, this is an excellent suggestion. To fully address the comment, a direct comparison with the direct deoxyfluorination of the corresponding tetralone (ketone) has been performed and the information has been added to the supporting information (please see below). The conventional deoxyfluorination approach using DAST delivers the product in a very low yield (18% by ¹⁹F NMR) of the desired tetralin and the conditions lead to significant substrate degradation. By comparison, our reaction conditions compare very favourably. We hope that this fully addresses the question from the referee.

The section in the supporting information reads as follows.

*"DAST (Diethylaminosulfur trifluoride) has been used in the deoxyfluorination of tetralone **S15** and this enables the formation of the difluorinated tetralin derivative **S16** in a moderate 38% yield.¹ However, the synthesis of difluorinated aryl tetralin **3a** from aryl tetralin **S17** has not been described yet. Using common deoxyfluorinating conditions,² we have demonstrated that the desired product **3a** can be obtained in a yield of 18% ¹⁹F NMR (see Figure S1). Full conversion, and substantial degradation, of the starting material **S17** was observed by TLC and ¹H*

¹ Adcock, A., Das Gupta, B., Khor, T.-C. Substituent effects by ¹⁹F nuclear magnetic resonance: Polar and π -electron effects. *Aust. J. Chem.* **29**, 2571-2581 (1976).

² Banerjee, A., Sudan Maji, M. A Brønsted Acid Catalyzed Cascade Reaction for the Conversion of Indoles to α -(3-Indolyl) Ketones by Using 2-Benzyloxy Aldehydes. *Chem. Eur. J.* **25**, 11521-11527 (2019).

NMR. In comparison, this approach allows for the synthesis of this complex motif **3a** in 57% isolated yield starting from the corresponding cyclobutanol **1a**.

Scheme S2: Comparison of different synthetic methodologies for the synthesis of difluorinated tetralin derivatives.

Figure S1: ^{19}F crude NMR of the deoxofluorination of **S17**. Ethyl 2-fluoroacetate was used as the internal standard.

The functional group tolerance of the current method is moderate (mostly halogens) but yields are good considering the one-pot sequence of three chemical transformations. A mention of what functional groups are not tolerated in this reaction would be helpful to assess its limitations. Nonetheless, the authors demonstrate its use for the synthesis of a derivative of nafenopin, which showcases the potential use of this method. Of note, the Supporting Information file is nicely organized and presented. The amount and quality of the NMR characterization data is impressive, and the authors are commended for their work.

Author Response: We appreciate the comments from the referee regarding the quality of the ESI and the synthetically useful levels of efficiency. In evaluating the scope, we focussed heavily on halogens and triflates to ensure that the final products could be easily modified by cross coupling reactions. The only potential limitation of the method is with highly electron-rich arenes and this is due to the (known) competing fluorination of the ring. For an example, please see Olah and co-workers, *Isr. J. Chem.* **39**, 207-210 (1999).

This paper should be publishable in Nature Communications after minor revisions. Of particular importance, a proposed mechanism with detailed intermediate structures (including rearrangement steps) is needed, either as the reaction design scheme or as a final scheme. This reviewer is particularly skeptical about the proposed nature of the cyclobutyl carbocation intermediate and why it would lead to a homoallylic fluoride product. Creary (*J. Org. Chem.* **2020**, *85*, 7086-7096.) has shown that when stabilized cyclobutyl electrophiles are ionized, cyclobutyl and bicyclobutonium cations are formed but eventually provide cyclobutane/cyclobutene products, not homoallylic products.

On the other hand, Champagne (10.26434/chemrxiv-2022-h9bq3-v2) has shown that in the Marek system cyclopropylcarbanyl cations are formed and can lead to homoallylic products directly or through homoallylic cations. The latter were always higher in energy, even when the homoallylic position is tertiary and benzylic. As such, this reviewer would hypothesize that in the current system, ionization of cyclobutanols **1** eventually leads to a cyclopropylcarbanyl cation (either directly or through a cyclobutyl/bicyclobutonium intermediate), which would explain the formation of the homoallylic fluoride **III** without requiring the homoallylic cation **II**. Some discussion about these possibilities is lacking here.

Author Response: We thank the referee most sincerely for this very insightful question and for investing so much time in considering the mechanistic postulate. Whilst we fully appreciate the alternatives suggestions that are raised, many of which we considered early in the work, we believe that the ultimate support for this mechanistic proposal is the regioselectivity of the reaction which manifests itself in the product aryl tetralin. A single regioisomer is always formed and this is completely independent of the electronic nature of the two aryl substituents. The initial cation formation is interesting on account of the 1,3-diaryl substitution and, to the best of our knowledge, such species have never been investigated computationally. If a bicyclobutonium cation was indeed generated, one might reasonably expect that the *pseudo*-symmetric nature of the intermediate would lead to a mixture of products. This is not the case.

The seminal work by Creary (*J. Org. Chem.* **2020**, *85*, 7086-7096) does not study 1,3-diaryl substitution but it does indicate that mono-aryl-substituted systems favour addition of exogenous nucleophiles. We have recently shown that under closely similar reaction condition, bicyclobutanes form cyclobutene derivatives (via cyclobutyl cations) and that fluoride can reversibly add (*ACS Catal.* **2022**, *12*, 14507–14516 – please see the top left hand side of the image shown below). However, ring contraction to form a cyclopropane only occurs in the presence of the catalysts. In this current study, we demonstrate that the initial homoallylic fluoride formation occurs by exposure to HF.

^aYields determined by ¹H and ¹⁹F NMR analyses.

Regarding the relative stabilities of the cations proposed, the cyclobutyl cation is tertiary but the aryl ring is twisted to minimise destabilising-non-covalent interactions (1,3-allylic strain). We therefore propose that rearrangement to form the secondary benzylic cation, in which the stabilising effect of the neighbouring π -system is not impeded, is reasonable and accounts for the selective formation of the intermediate.

To further support our hypothesis, we have attained an X-ray structure of the carbinol starting material (in this case **minor-1c** (*trans*), please see below) that shows that the aryl ring is rotated out of place. Whilst this is not directly comparable with the cation, in both cases 1,3-allylic strain will position the plane of the phenyl ring orthogonal to the empty p -orbital thereby minimising any stabilising effect. Rearrangement induced by steric decompression generates the secondary benzylic cation in which the stabilising influence of the aryl group is not impeded by the cyclic scaffold. We believe that these resonance structures provide support for the experimentally observed regioselectivity of the transformation and the insensitivity of the process to changes in the electronic nature of the aryl ring. The latter point speaks against (pseudo)-symmetric bicyclobutonium-like species.

The most compelling evidence that speaks against the formation of a transient cyclopropylcarbinyl cation stems from the work of Li and co-workers (*Org. Lett.* **2021**, *23*, 3088-3093; please see below). The authors demonstrate that exposure of aryl-substituted methylenecyclopropanes to Selectfluor[®] and HF (i.e. uncatalysed like the initial

steps of our cascade), trigger a Wagner-Meerwein rearrangement to form a difluorocyclobutane. The cyclopropylcarbanyl cation that is formed does not lead to the homoallylic fluoride observed in our study.

gem-Difluorination of Methylene-cyclopropanes (MCPs) Featuring a Wagner–Meerwein Rearrangement: Synthesis of 2-Arylsubstituted gem-Difluorocyclobutanes

Peng-Peng Lin, Long-Ling Huang, Si-Xin Feng, Shuang Yang, Honggen Wang, Zhi-Shu Huang, and Qingqiang Li*

Cite This: *Org. Lett.* 2021, 23, 3088–3093

Read Online

ACCESS |

Metrica & More

Article Recommendations

Supporting Information

ABSTRACT: The geminal difluorocyclobutane core is a valuable structural element in medicinal chemistry. Strategies for gem-difluorocyclobutanes, especially the 2-substituted cases, are limiting and often suffer from harsh reaction conditions. Reported herein is a migratory gem-difluorination of aryl-substituted methylene-cyclopropanes (MCPs) for the synthesis of 2-arylsubstituted gem-difluorocyclobutanes. Commercially available Selectfluor (F-TEDA-BF₄) and Py-HF were used as the fluorine sources. The protocol proceeds via a Wagner–Meerwein rearrangement with mild reaction conditions, good functional group tolerance, and moderate to good yields. The product could be readily transformed to gem-difluorocyclobutane-containing carboxylic acid, amine, and alcohol, all of which are useful building blocks for biologically active molecule synthesis.

The elegant work by Champagne on the venerable Marek-type cations (10.26434/chemrxiv-2022-h9bq3-v2) is certainly very interesting but cation generation is never investigated starting from ionisation of the cyclobutanol, but rather from the cyclopropanol. It is also important to note that this generates an (internal) 1,2-disubstituted alkene and not the 1,1-disubstituted alkene that is required for the catalysed difluorination / rearrangement step. Under these reaction conditions, 1,2-disubstituted alkenes are recalcitrant to I(I/III)-mediated difluorination. In the earlier work by Marek (*J. Am. Chem. Soc.* 2020, 142, 12, 5543–5548), the authors also report that the alkene motif of the product homoallylic fluoride is 1,2-disubstituted and not 1,1- as observed here.

We hope that these comments address the referee's sufficiently and appreciate the very thoughtful mechanistic questions. The referee is quite right that a discussion of the factors that led to our mechanistic hypothesis would enhance the text and so we have added the following statements to the paper, which cite the important studies that the referee has highlighted.

Starting on p2, line 18:

This would ultimately complement the elegant studies by Lanke and Marek on the generation of trans-1,2-disubstituted homoallylic fluorides, via cyclopropinyl carbocations, from cyclopropyl carbinols.[27] Although the well-documented involvement of cyclopropinyl[27,28] and bicyclobutonium cations cannot be completely excluded[29], studies from Li and co-workers have demonstrated that exposure of aryl-substituted methylene cyclopropanes to Selectfluor® and HF trigger a Wagner-Meerwein rearrangement to generate difluorocyclobutanes[30] and do not yield homoallylic fluorides. Further confidence in this approach stemmed from the expected 1,3-allylic strain[31] in the diaryl cyclobutyl cation that would prevent the aryl ring from being co-planar with the p-orbital. Steric decompression alleviates this scenario, ultimately leading to the formation of the 1,1-disubstituted alkene: this species can then engaged in an I(I)/I(III) catalysis cycle,[32] enabling a regioselective 1,1-difluorination[33-40] to occur via a precedented phenonium ion rearrangement.[41]

Starting on p4, line 17:

“It is pertinent to mention that crystals of the trans isomer of the starting cyclobutanol (minor-1c) could be isolated and subjected to X-ray diffraction analysis (see Table 1 legend).[46] Although comparisons with the

cyclobutyl carbocation should be drawn with caution, it is clear that the aryl ring next to ionisation site is twisted out of plane in order to minimise 1,3-allylic strain.

Starting on p8, line 7:

"It is interesting to note that the transformation is characterised by formation of a single regioisomer, which further supports the mechanistic framework in Fig. 1 and is difficult to reconcile with the formation of a bicyclobutonium intermediate."

Minor comments:

- Compounds are all racemic, could it be indicated as so in the schemes?

Author Response: This is an excellent suggestion. The manuscript has been modified accordingly and the "±" sign has been added to denote that the products are racemic.

- That Olah's reagent is Py•HF is not indicated anywhere in the manuscript. Also, how the amine•HF mixtures are made using Et₃N•3HF and Py•HF should be discussed somewhere in the text.

Author Response: This information is provided in the supporting information with a full description. An additional reference to the ESI was added to the text.

Starting on p4, line 16:

"(see ESI for further information on the preparation of amine:HF mixtures)"

- The impact of solvent quantities should probably be discussed as a function of solute concentration rather than solvent volume (0.5 – 2mL), to make it more general.

Author Response: This is an excellent suggestion but we respectfully feel that the way in which the data is currently presented is more accessible and understandable for practitioners.

REVIEWER COMMENTS

Reviewer #1 (Remarks to the Author):

The authors have performed the necessary corrections.

There are some minor corrections that need to be incorporated in the ESI.

The D...A distances must be reported with esd's for all the non-covalent contacts involving fluorine atoms.

The H...A distances must be rounded off to 2 decimal places, DHA angles to the nearest integer and esd's must be present if the hydrogens have been located and refined for D-H, H...A distance and DHA angle. If fixed no esd's must be mentioned.

A sentence on how the hydrogens were treated during refinement can be mentioned in the section on X-ray crystal structure analysis for the compounds reported.

Following these corrections, I recommend publication in Nat.Comm.

The authors acknowledged the largest issue from my report, which is the identity of their proposed cationic intermediates, but have not addressed it satisfactorily. It is clear to this reviewer that the homoallylic fluoride compound can be accessed from either the cyclobutyl, cyclopropylcarbiny, or homoallylic cation intermediate (see below). In order to provide support to my review, I obtained DFT-optimized structures of the relevant cations and present them below. I don't suggest that the authors perform a full-scale computational study, but that they at least consider the true structures in the proposed mechanism.

First, the issue of the cyclobutyl cation should be addressed. As seen in structure **A** below, it is a slightly puckered cyclobutyl cation, with the aryl substituent perfectly aligned with the cation's p orbital. As such, the claim of 1,3-allylic strain that the authors put forward in the introduction (page 2 line 24) is simply incorrect. So is the discussion around the crystal structure of the minor-**1c** (page 4, line 20), since the structure of the alcohol substrate is absolutely irrelevant with regards to the cation structure. Any mention to this 1,3-allylic strain hypothesis should thus be removed.

Second, structure **B** is not a fully closed cyclopropylcarbiny, nor a fully open homoallylic structure, but rather in between. It is also around 7.0 kcal/mol higher in energy than **A**. As the transition structures for nucleophilic attack will resemble the cationic intermediates, it is likely (but not obligatory) that nucleophilic attack on **A** will have a lower activation energy. Figuring out these pathways would require a full study which is not the objective of this article and I don't ask that the authors perform that. Overall,

this preliminary data matches with the known literature that indicates that homoallylic cations are high in energy, relative to cyclopropylcarbinyl or cyclobutyl/bicyclobutonium structures. The authors should not use that data in their paper, but their proposal should discuss the possibilities, and especially the known instability of homoallylic cations.

In addition, I believe the authors have misinterpreted (discussed on page 2, line 22) and that section of the paper should be rephrased. Li's reaction involves a cyclopropylcarbinyl cation that is presumed to prefer the cyclobutyl/bicyclobutonium form due to the fluorine's stabilization of α -cations. This cyclobutyl cation is what presumably leads to a cyclobutyl product. This is similar to Creary's work, where a cyclobutyl cation leads to a cyclobutyl product. In the current study a homoallylic cation is formed but as shown above it looks like the cyclobutyl cation is more stable, so this interesting fact needs to be contrasted to previous work and compared to the Marek reaction.

I would also disagree with the authors' claim on page 8, line 7 that the formation of a single regioisomer is due to a specific cation (also in the rebuttal letter). Even in complex cyclopropylcarbinyl/bicyclobutonium cations, one nucleophilic attack is usually much more favored energetically, so even a bridged structure could lead to the homoallylic fluoride specifically (see Marek's paper). As was shown above, this experimental observation does not need a homoallylic cation as the major intermediate, but the intermediate that leads to their product might be a cyclopropylcarbinyl/homoallyl hybrid structure.

Finally, the authors name the cyclopropylcarbinyl cations as "cyclopropinyl" in a few instances, which should be corrected. Once these issues are addressed, the manuscript should be publishable in Nature Communications since the quality and importance of the results remain high.

REVIEWER COMMENTS

- **Reviewer #1 (Remarks to the Author):**

The authors have performed the necessary corrections.

There are some minor corrections that need to be incorporated in the ESI.

The D...A distances must be reported with esd's for all the non-covalent contacts involving fluorine atoms. The H...A distances must be rounded off to 2 decimal places, DHA angles to the nearest integer and esd's must be present if the hydrogens have been located and refined for D-H, H...A distance and DHA angle. If fixed no esd's must be mentioned.

A sentence on how the hydrogens were treated during refinement can be mentioned in the section on X-ray crystal structure analysis for the compounds reported.

Following these corrections, I recommend publication in Nat.Comm.

Author response: The additional information has been added to the ESI. We thank the referee for the helpful comments.

- **Reviewer #3 (Remarks to the Author):**

The authors acknowledged the largest issue from my report, which is the identity of their proposed cationic intermediates, but have not addressed it satisfactorily. It is clear to this reviewer that the homoallylic fluoride compound can be accessed from either the cyclobutyl, cyclopropylcarbinyl, or homoallyl cation intermediate (see below). In order to provide support to my review, I obtained DFT-optimized structures of the relevant cations and present them below. I don't suggest that the authors perform a full-scale computational study, but that they at least consider the true structures in the proposed mechanism.

Author response: The postulated reaction mechanism has been modified to include two of the cationic intermediates that the referee has suggested.

First, the issue of the cyclobutyl cation should be addressed. As seen in structure **A** below, it is a slightly puckered cyclobutyl cation, with the aryl substituent perfectly aligned with the cation's p orbital. As such, the claim of 1,3-allylic strain that the authors put forward in the introduction (page 2 line 24) is simply incorrect. So is the discussion around the crystal structure of the minor-**1c** (page 4, line 20), since the structure of the alcohol substrate is absolutely irrelevant with regards to the cation structure. Any mention to this 1,3-allylic strain hypothesis should thus be removed.

Author response: We have removed the reference and the discussion regarding 1,3-allylic strain throughout the manuscript following the referee's suggestion.

Second, structure **B** is not a fully closed cyclopropylcarbinyll, nor a fully open homoallylic structure, but rather in between. It is also around 7.0 kcal/mol higher in energy than **A**. As the transition structures for nucleophilic attack will resemble the cationic intermediates, it is likely (but not obligatory) that nucleophilic attack on **A** will have a lower activation energy. Figuring out these pathways would require a full study which is not the objective of this article and I don't ask that the authors perform that. Overall,

this preliminary data matches with the known literature that indicates that homoallylic cations are high in energy, relative to cyclopropylcarbinyll or cyclobutyl/bicyclobutonium structures. The authors should not use that data in their paper, but their proposal should discuss the possibilities, and especially the known instability of homoallylic cations.

Author response: The working hypothesis has been modified to show these possibilities that the referee describes. Specifically, a dashed line has been to intermediate **II** and the possibility of direct attack at the cyclobutonium cation has been discussed. The referee has indicated that this data should not be referred to in the paper, nor should a full computational study be performed.

In addition, I believe the authors have misinterpreted the Li study (discussed on page 2, line 22) and that section of the paper should rephrased. Li's reaction involves a cyclopropylcarbinyll cation that is presumed to prefer the cyclobutyl/bicyclobutonium form due to the fluorine's stabilization of α -cations. This cyclobutyl cation is what presumably leads to a cyclobutyl product. This is similar to Creary's work, where a cyclobutyl cation leads to a cyclobutyl product. In the current study a homoallylic fluoride is formed but as shown above it looks like the cyclobutyl cation is more stable, so this interesting fact needs to be contrasted to previous work and compared to the Marek reaction.

Author response: We have modified the descriptions of Li's study as suggested by the referee.

I would also disagree with the authors' claim on page 8, line 7 that the formation of a single regioisomer is due to a specific cation (also in the rebuttal letter). Even in complex cyclopropylcarbinyll/bicyclobutonium cations, one nucleophilic attack is usually much more favored energetically, so even a bridged structure could lead to the homoallylic fluoride specifically (see Marek's paper). As was shown above, this experimental observation does not need a homoallylic cation as the major intermediate, but the intermediate that leads to their product might be a cyclopropylcarbinyll/homoallyll hybrid structure.

Author response: We have removed this statement following the referee's suggestion.

Finally, the authors name the cyclopropylcarbinyll cations as "cyclopropinyll" in a few instances, which should be corrected. Once these issues are addressed, the manuscript should be publishable in Nature Communications since the quality and importance of the results remain high.

Author response: This has been corrected. We thank the referee for the very helpful and insightful comments.